# The Efficient Design of Lossy P-LDPC Codes over AWGN Channels

**Runfeng Wang** [1] , **Sanya Liu** [2], **Huihui Wu** [3] **and Lin Wang** [1,*]

1    Department of Information and Communication Engineering, Xiamen University, Xiamen 361005, China
2    Xiamen Key Laboratory of Mobile Multimedia Communications, College of Information Science and Engineering, Huaqiao University, Xiamen 361021, China
3    Department of Electrical and Computer Engineering, McGill University, Montreal, QC H3A 0G4, Canada
*    Correspondence: wanglin@xmu.edu.cn

**Abstract:** Considering the high compression requirements of transmission, lossy block codes are particularly concerned due to their good compression performance and simple implementation. This paper investigates and analyzes the distortion rate performance of protograph LDPC (P-LDPC) codes for Bernoulli sources over AWGN channels. We first analytically establish the connection between the parity check matrix of a P-LDPC code and the extra distortion caused by the noisy channels. It was found that the additional distortion related to channel noise increases with the rising total degree of a parity check matrix. Further, two design algorithms are proposed for optimizing lossy multirate P-LDPC codes, considering the effect of noisy channels. Finally, simulation results demonstrate the robustness of the optimized P-LDPC codes over noisy channels.

**Keywords:** lossy compression; protograph LDPC codes; quantization; distortion rate; AWGN channels





## 1. Introduction

Efficient compression can save hardware storage resources and reduce bandwidth consumption during transmission. Compared with lossless compression, lossy compression can compress data to a greater extent, resulting in a rate smaller than the entropy of the source. In the practical transmission environment, the compressed bits may be interfered by the channel noise, leading to extra transmission distortion, and thus, the study of lossy compression over noisy channels is of great significance.

Lossy compression is based on the distortion rate theory [1,2], and the source is quantized by source coding. Source coding and channel coding have a duality [3], so source compression can be realized by channel coding. Quantization using channel codes is to replace the equivalent codes of the quantization subspace [4] by the channel codes. Therefore, it is of great significance to find error-correcting codes with low complexity and suitable for different source characteristics. Low-density parity check (LDPC) codes are found to be feasible for lossy compression [5], and they have been proven to asymptotically achieve the distortion rate limit for the equal-probability binary symmetric sources, but no specific codes have been proposed [6]. As the dual codes of LDPC codes, low-density generator matrix (LDGM) codes are proposed to compress the Bernoulli sources, which has made the distortion close to the distortion rate limit [7,8]. As a subclass of LDPC codes, protograph LDPC (P-LDPC) codes can obtain their parity check matrix (*H*) by extending a small protomatrix [9,10]. Compared with the randomly generated LDPC codes, P-LDPC codes have linear encoding and decoding complexity. Therefore, the lossy compression using P-LDPC codes has lower complexity, which makes it feasible in practical scenarios. For the compression of Bernoulli sources, Liu [11] proposed two algorithms to re-design P-LDPC codes over noise-free channels. The resultant codes achieved a compression performance close to the distortion rate limit. However, in practical applications, a complete communication system has to consider the sensitivity to noisy channels.

For the compression procedure, the P-LDPC codes and the reinforced belief propagation (RBP) algorithm [12] jointly operate on the sources, serving as a quantization [13,14] scheme for binary sources. The size of the quantization subspace of P-LDPC codes is related to its error correction ability. Since the degree of each variable node of a P-LDPC code varies, the error correction ability of each variable node is different, and this causes the non-uniform error correction interval (i.e., quantized subspace). Therefore, when P-LDPC codes are used for source compression, we need to re-design the P-LDPC codebooks to minimize the quantization distortion ($D_S$). Further, the index assigned to each codeword after quantization is transmitted over the channels, and the corrupted index will lead to extra channel distortion ($D_C$) [15–17]. In this paper, we view information bits of each codeword as an index of the linear block codes. Moreover, it is noted that the design of lossy P-LDPC codes only concerns the parity check matrices, taking into account the index assignment and the noisy channels. We point out that the RBP algorithm for lossy P-LDPC codes and the standard BP algorithm for channel LDPC codes could share the same decoding module on a ship, improving the on-ship area efficiency.

For underwater acoustic communications or power line communications, the robustness of a compression scheme to harsh channel conditions has to be considered, since the channel distortion $D_C$ could become the dominant distortion. Therefore, we first analyze the key factors affecting the channel distortion and then consider the influence of channel noise while optimizing the P-LDPC codes. By doing so, the average distortion at both low and high signal-noise ratio (SNR) regions could be balanced. The main contributions of this paper are described as follows:

1. The connection between $D_C$ and the total degree of the variable nodes of a protomatrix is derived and experimentally verified.
2. Two optimization algorithms are proposed based on the principle of balanced degree allocation, and the optimized codes have excellent compression performance and error resilience.

The remainder of this paper is organized as follows. Section 2 introduces the system model. The optimization of the P-LDPC codes is presented in Section 3, including the analysis for the low SNR region and optimization for the high SNR region. The experimental results and analysis are given in Section 4. The conclusion is given in Section 5.

## 2. System Model

The Bernoulli source compression system model using P-LDPC codes as the quantization codebook over noisy channels is shown in Figure 1.

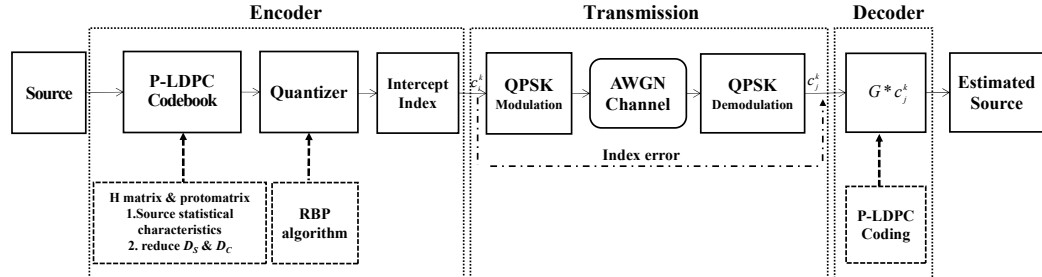

**Figure 1.** Lossy compression transmission system using P-LDPC codes over AWGN channels for Bernoulli sources.

$S^N$ represents the Bernoulli source, and $s^N = (s_0, s_1, \cdots, s_{N-1})^T$ is a Bernoulli source sequence of length $N$. $c^N$ is the quantized codeword. We used the Hamming distance to measure the distortion, i.e.,

$$d_H(s_i, \hat{s}_i) = \begin{cases} 0 & if \ s_i = \hat{s}_i \\ 1 & if \ s_i \neq \hat{s}_i, \end{cases} \tag{1}$$

$$d(s^N, \hat{s}^N) = \frac{1}{N} \sum_{i=0}^{N-1} d_H(s_i, \hat{s}_i). \tag{2}$$

For the compression system using linear block codes, $k$ is the length of information bits, which is also the length of the binary index $c^k$ of the P-LDPC codes. The distortion $D_S$ and $D_C$ are expressed, respectively, by:

$$D_S = \frac{1}{N} \sum_{i=0}^{2^k-1} \sum_{j=0}^{2^{N-k}-1} p\left(s_{ij}^N\right) d\left(s_{ij}^N, c_i^N\right), \tag{3}$$

$$D_C = \frac{1}{N} \sum_{i=0}^{2^k-1} \sum_{j=0}^{2^k-1} P\left(c_i^k\right) P\left(c_j^k \middle| c_i^k\right) d\left(c_i^N, c_j^N\right), \tag{4}$$

where $s_{ij}^N$ is the $j$th source sequence of the $i$th quantization subspace. As pointed out in [15], when the quantized codeword and subspace division satisfy the centroid codeword and minimum distortion division, the total distortion is equal to the sum of $D_S$ and $D_C$.

This system consists of three parts. The first part is the encoding module. This module is the quantization module, which introduces the quantization distortion $D_S$. In this module, the parity check matrix $H$ of the P-LDPC codes is designed by considering the statistical characteristics of the source, so as to reduce $D_S$ as much as possible. Then, by considering the influence of channel noise on the index, combined with the corresponding relationship between the index and the codeword, constraints are added to the $H$ matrix to reduce $D_C$. Next, by using the RBP algorithm, each quantization subspace of the source is shrunk to a codeword of P-LDPC codes, so as to achieve the purpose of lossy compression. The schematic diagram is shown in Figure 2, which is to quantize the $N$-bit Bernoulli source $S^N$ to $2^k$ P-LDPC codewords $c^N$, and the quantization subspace (coding subspace) is the error correction range of the LDPC codes, then through the interception of information bits, that is to extract the index for transmission, to further compress the source.

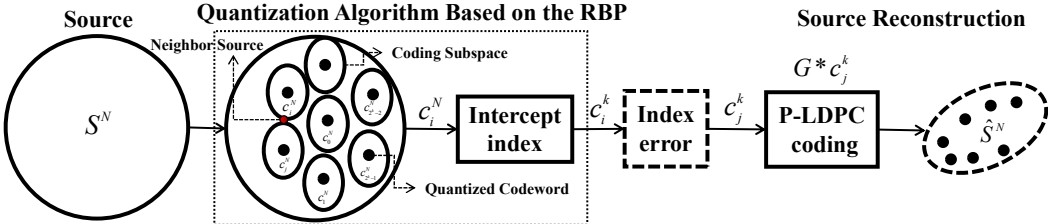

**Figure 2.** Simplified diagram of system principle.

The second part is the transmission module. In this module, the $k$-bit binary index is modulated by QPSK, and after transmitting over AWGN channels, QPSK demodulation is used to obtain the index with possible errors. The third part is the decoding module, where the generator matrix ($G$) corresponding to the $H$ matrix is multiplied by the demodulated index to perform linear decoding.

## 3. Analysis and Optimization Methods

### 3.1. Analysis for Low SNR Region

For the low SNR region, channel noise will cause serious errors on the index during transmission, leading to a large $D_C$. We can minimize $D_C$ by reducing the distance between the two quantization subspaces corresponding to adjacent indices of the P-LDPC codes.

Now, let us assume $n_1$ is an original source sequence of length $N$, $n_2$ is a reconstructed sequence at the receiver, and $w$ is the Hamming distance between the two sequences. Moreover, let $u_1$ represent the $k$-bit binary index, which is a short sequence transmitted

over the channels. Let $\boldsymbol{u}_2$ be the short sequence demodulated after demodulation. To this end, $D_C$ can be expressed by

$$
\begin{aligned}
D_C &= w \\
&= \|\boldsymbol{n}_1 \oplus \boldsymbol{n}_2\|_1 \\
&= \|\boldsymbol{G}(\boldsymbol{u}_1 \oplus \boldsymbol{u}_2)\|_1 .
\end{aligned}
\tag{5}
$$

First, suppose that a one-bit error occurs in the binary index during transmission, and let $\boldsymbol{H}_{m \times n} = [\boldsymbol{H}_{m \times k} | \boldsymbol{I}_{m \times m}]$ be a systematic matrix. Without loss of generality, we assume the single-bit error occurs at the last position of the index, and then, we have

$$
\begin{aligned}
\Delta \boldsymbol{u} &= \boldsymbol{u}_1 \oplus \boldsymbol{u}_2 \\
&= (0, 0, \cdots, 0, 1)^T ,
\end{aligned}
\tag{6}
$$

$$
\begin{aligned}
\boldsymbol{G} \cdot \Delta \boldsymbol{u} &= \begin{bmatrix} \boldsymbol{I}_{k \times k} \\ \boldsymbol{H}_{m \times k} \end{bmatrix} \cdot \Delta \boldsymbol{u} \\
&= \left( \overbrace{0, 0, \cdots, 0, 1}^{k}, \overbrace{h_{1k}, h_{2k} \cdots, h_{mk}}^{m} \right)^T ,
\end{aligned}
\tag{7}
$$

where $h_{ij}$ is the $(i, j)$th entry of the $\boldsymbol{H}_{m \times n}$ matrix and $i \in [1, m]^*$ and $j \in [1, k]^*$, where $*$ represents integers in this range. Therefore, the source distortion generated by the one-bit error of the binary index is:

$$
\begin{aligned}
w_1 &= 1 + \sum_{i=1}^{m} h_{ik} \\
&= 1 + V(k),
\end{aligned}
\tag{8}
$$

where $V(k)$ is the degree of the $k$th variable node. Therefore, the average distortion (in bits) generated by the one-bit error of the binary index is:

$$
\begin{aligned}
\overline{w}_1 &= \frac{1}{k} \left( k + \sum_{j=1}^{k} \sum_{i=1}^{m} h_{ij} \right) \\
&= 1 + \frac{1}{k} \sum_{j=1}^{k} V(j).
\end{aligned}
\tag{9}
$$

Following the same way, we assume a two-bit error occurs in the index, and the corresponding distortion becomes

$$
w_2 = 2 + V(j) + V(\hat{j}),
\tag{10}
$$

where $j \neq \hat{j}$ and the average distortion in bits is:

$$
\begin{aligned}
\overline{w}_2 &= 2 + \frac{1}{C_k^2} \sum_{\hat{j}=1}^{k} \left[ V(\hat{j}) + \sum_{j \neq \hat{j}}^{k} V(j) \right] \\
&= 2 + \frac{k-1}{C_k^2} \sum_{j=1}^{k} V(j) \\
&= 2 \left[ 1 + \frac{1}{k} \sum_{j=1}^{k} V(j) \right].
\end{aligned}
\tag{11}
$$

Finally, the average distortion generated by an $e$-bit error of the binary index is:

$$\overline{w}_e = e \cdot \left[ 1 + \frac{1}{k} \sum_{j=1}^{k} V(j) \right] = \frac{e}{k} V(\boldsymbol{G}), \tag{12}$$

where $V(\boldsymbol{G})$ is the total degree of the $\boldsymbol{G}$ matrix. The check matrix of P-LDPC codes whose copy times are $Z$ fills the cyclic matrix in the $M * N$ square matrices after the protomatrix is copied. Therefore, for P-LDPC codes, the distortion formula of the $e$-bit error of the binary index can be changed to:

$$\begin{aligned} \overline{w}_e &= e \cdot \left[ 1 + \frac{1}{(N-M) \cdot Z} \sum_{j=1}^{N-M} \sum_{i=1}^{M} \left( Z \cdot B_{ij} \right) \right] \\ &= e \cdot \left[ 1 + \frac{1}{N-M} \sum_{j=1}^{N-M} V(j) \right], \end{aligned} \tag{13}$$

where $B_{ij}$ is the $(i,j)$th element in the protomatrix $\boldsymbol{B}_{M \times N}$.

Figure 3 shows the relationship between $D_C$ (in bits) and the total degree of variable nodes in a $3 \times 5$ protomatrix with code rate 0.4, and the length of the information bits is 1000. As can be seen from Figure 3a, a small number of index errors have little effect on the compression system, showing a slow linear growth; a large number of index errors cause a sharp rise in $D_C$, while the proportion of the total degree of the $\boldsymbol{H}$ matrix to the $D_C$ influence becomes larger. After the number of assumed index error bits is fixed, we can obtain the simplified Figure 3b from Figure 3a. As can be seen from Figure 3b, if the index error increases (the channel condition is worse), the total degrees have more influence on $D_C$.

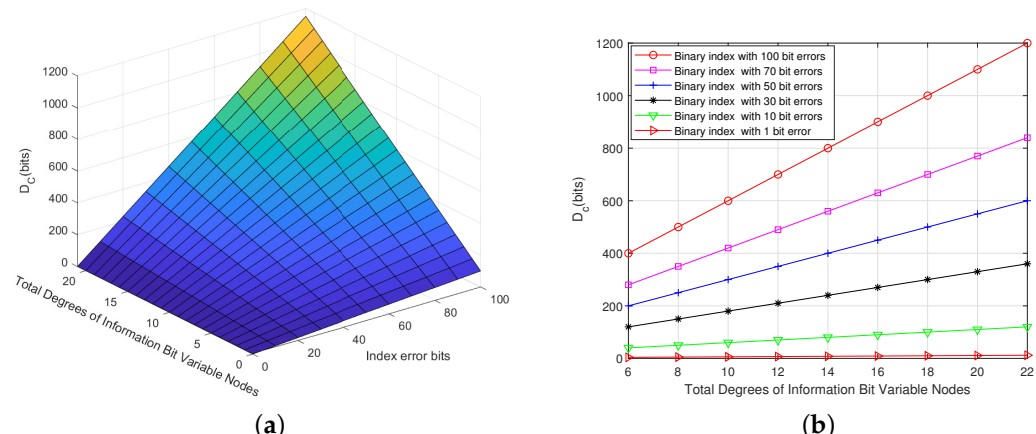

**Figure 3.** Relationship between total degrees corresponding to information bits in the protomatrix and $D_C$. (**a**) Effects of different transmission errors and total degrees on $D_C$ (**b**) Distortion effects of increased total degrees for different index errors.

For continuous burst errors that may occur in other channels, we now assume that the probability of two consecutive errors during transmission is independent and identical, and then, we assume the two-bit error occurs at the last two positions of the index, i.e.,

$$\Delta \boldsymbol{u} = \boldsymbol{u}_1 \oplus \boldsymbol{u}_2 = (0, 0, \cdots, 1, 1)^T. \tag{14}$$

The source distortion in this case is:

$$\begin{aligned} w_{\tilde{2}} &= 2 + \sum_{i=1}^{m} h_{i(k-1)} + \sum_{i=1}^{m} h_{ik} \\ &= 2 + V(k-1) + V(k). \end{aligned} \tag{15}$$

The different short sequence indices transmitted are connected in the channels, and each index bit transmitted affects each bit variable node on the information bit. Let $V_k^0$ represent the $k$th position error of the 0th index short sequence, and the $D_C$ caused by it is affected by the degree of the variable node at the $k$th position in the $\boldsymbol{H}$ matrix:

$$
\overbrace{\cdots V_{k-2}^0 \; V_{k-1}^0 \; V_k^0}^{u_0} \; \overbrace{V_1^1 \; V_2^1 \cdots V_k^1}^{u_1} \; \overbrace{V_1^2 \; V_2^2 \; V_3^2 \cdots}^{u_2} \; . \tag{16}
$$

When the index has two consecutive errors in transmission, the corresponding variable nodes that affect the source distortion are (assuming that there can be consecutive errors at the beginning and end)

$$
\begin{array}{c}
\cdots V_k^0 \; \boxed{V_1^1} \\[2pt]
\boxed{V_1^1 \; V_2^1} \\[2pt]
\cdots\cdots \\[2pt]
\boxed{V_{k-1}^1 \; V_k^1} \\[2pt]
\boxed{V_k^1} \; V_1^2 \cdots .
\end{array} \tag{17}
$$

Taking all possible errors to calculate the average channel distortion of a source sequence, the average distortion (in bits) caused by two and three consecutive index errors is, respectively:

$$
\overline{w}_{\tilde{2}} = \frac{2}{k+1}\left[ k + \sum_{j=1}^k V(j) \right], \tag{18}
$$

$$
\overline{w}_{\tilde{3}} = \frac{3}{k+2}\left[ k + \sum_{j=1}^k V(j) \right]. \tag{19}
$$

Therefore, the average distortion caused by $e$ consecutive index errors is:

$$
\overline{w}_{\tilde{e}} = \frac{e}{k+e-1}\left[ k + \sum_{j=1}^k V(j) \right]. \tag{20}
$$

A special case when $e = 1$ and $k \to \infty$ is:

$$
\begin{aligned}
&\overline{w}_{\tilde{1}} = \overline{w}_1 = V(\boldsymbol{G})/k, \\
&\lim_{k \to \infty} \overline{w}_{\tilde{e}} = \lim_{k \to \infty} \overline{w}_e = e.
\end{aligned} \tag{21}
$$

It can be summarized from the aforementioned analysis that $D_C$ is positively related to the total degree of the parity check matrix $\boldsymbol{H}$. Therefore, a low-weight parity check matrix is highly preferred for reducing $D_C$ in the low SNR regime.

### 3.2. Optimization for High SNR Region

It is known that $D_C$ would be small for the high SNR regime, and thus, more attention should be paid to reducing source distortion $D_S$. In order to reduce $D_S$ caused by quantization, the division of the quantization subspace needs to be more suitable for source statistical characteristics, and the equivalent codeword of the quantization subspace needs to be closer to the central codeword. In the compression system using linear block codes as the quantization codebook, we can adjust the partition of the quantization subspace and quantization codebook by assigning the degree of variable nodes.

In order to do so, we propose a new algorithm designing the P-LDPC codes based on the codes of [11]. The proposed Algorithm 1 including the check degree allocation (CA) algorithm and the check-variable degree allocation (CVA) algorithm is based on the concept of degree balance, and the resultant codes achieve better distortion rate performance in the high SNR regime, as shown in the next section. In addition, the proposed Algorithm 2 is based on the global degree allocation (GDA) algorithm, which can find codes of better compression performance. Moreover, the GDA algorithm can also achieve a variable code rate and balance compression performance and channel noise robustness by selecting appropriate degrees. The notations in Algorithms 1 and 2 are explained as follows:

---

**Algorithm 1** CA algorithm and CVA algorithm

---

**Require:** AR3A-impr codes and AR4JA-impr codes [11]
**Ensure:** CA-AR3A-impr codes and CVA-AR3A-impr codes;
  CA-AR4JA-impr codes and CVA-AR4JA-impr codes.
 1: **for** $j = M + 1 \rightarrow N$ **do**
 2:   $C_{xj} \leftarrow V_j / M$
 3:   **if** $x \geq M - V_j \% M$ **then**
 4:     $C_{xj} \leftarrow C_{xj} + 1$
 5:   **else**
 6:     $C_{xj} \leftarrow C_{xj}$
 7:   **end if**
 8: **end for**
 9: **if** $\prod_{x=1}^{M} C_{xj} = \prod_{x=1}^{M} C_{xj}^{input}$ **then**
10:   $C_{xj} \leftarrow C_{xj}^{input}$
11: **else**
12:   $C_{xj} \leftarrow C_{xj}$
13: **end if**
     // End of the degree allocation of check nodes.
     // Output CA-AR3A-impr and CA-AR4JA-impr codes.
14: **for** $i = 1 \rightarrow M$ **do**
15:   $V_{iy} \leftarrow (C_i - 1)/(N - M)$
       // $y \geq M$
16:   **if** $y \geq N - (C_i - 1)\%(N - M)$ **then**
17:     $V_{iy} \leftarrow V_{iy} + 1$
18:   **else**
19:     $V_{iy} \leftarrow V_{iy}$
20:   **end if**
21: **end for**
     // End of the degree allocation of variable nodes.
22: **repeat**
23:   swap ( $V_{iy}, V_{iy'}$ )
24: **until**
$$\begin{cases} V_j = (\sum_{j=M+1}^{N} V_j)/(N-M), & if \quad M < j < N - (\sum_{j=M+1}^{N} V_j)\%(N-M) \\ V_j = (\sum_{j=M+1}^{N} V_j)/(N-M) + 1, & if \quad j \geq N - (\sum_{j=M+1}^{N} V_j)\%(N-M) \end{cases}$$
25: **return** CA-AR3A-impr codes; CA-AR4JA-impr codes; CVA-AR3A-impr codes; CVA-AR4JA-impr codes

---

$V_j$: the degree of the $j$th variable node ($j \in [M+1, N]^*$);
$C_i$: the degree of the $i$th check node ($i \in [1, M]^*$);
$d$: the total degree of the variable nodes associated with the information bit in the protomatrix;
$V_{iy}$: the number of edges connecting the $i$th check node and the $y$th variable node ($y \in [M+1, N]^*$);

$C_{xj}$: the number of edges connecting the $j$th variable node and the $x$th check node ($x \in [1, M]^*$).

Algorithm 1 optimizes the protomatrices with good compression performance based on the existing codes from [11] by reassigning the degrees of rows and columns. The following Algorithm 1 takes the AR3A-impr codes and the AR4JA-impr codes [11] as an example. Note that Algorithm 1 could be used for any codes of [11].

When the code rate is 0.57, the original and optimized codes are shown as in Equation (22).

$$
\begin{array}{c}
\begin{bmatrix} 1 & 0 & 0 & 0 & 0 & 1 & 2 \\ 0 & 1 & 0 & 1 & 2 & 1 & 3 \\ 0 & 0 & 1 & 2 & 1 & 2 & 2 \end{bmatrix} \rightarrow \begin{bmatrix} 1 & 0 & 0 & 1 & 1 & 1 & 2 \\ 0 & 1 & 0 & 1 & 1 & 1 & 3 \\ 0 & 0 & 1 & 1 & 1 & 2 & 2 \end{bmatrix} \\
\scriptstyle AR3A-impr \qquad\qquad\qquad CA-AR3A-impr \\
\downarrow \\
\begin{bmatrix} 1 & 0 & 0 & 1 & 1 & 1 & 2 \\ 0 & 1 & 0 & 1 & 2 & 2 & 1 \\ 0 & 0 & 1 & 2 & 1 & 1 & 2 \end{bmatrix} \leftarrow \begin{bmatrix} 1 & 0 & 0 & 1 & 1 & 1 & 2 \\ 0 & 1 & 0 & 1 & 1 & 2 & 2 \\ 0 & 0 & 1 & 1 & 1 & 2 & 2 \end{bmatrix} . \\
\scriptstyle CVA-AR3A-impr \qquad\qquad\qquad CVA-AR3A-impr-temp
\end{array} \tag{22}
$$

Algorithm 2 (GDA) optimizes the P-LDPC codes from scratch and uses the principle of degree balance to generate new codes suitable for equal-probability Bernoulli sources by global degree allocation. The initial structure of the proposed P-LDPC codes is shown in Equation (23), which could make the partition of the quantization subspace by the P-LDPC codes concentrated in $B_{M \times k}$. Algorithm 2 can make the error correction range of each variable node as equal as possible by filling the total degree into $B_{M \times k}$, so that $2^k$ quantized subspaces are further uniformly divided. By doing so, the quantization distortion $D_S$ is reduced. In addition, P-LDPC codes with various rates could be obtained from Algorithm 2, by arbitrarily setting the dimension of the identity matrix and the number of variable nodes in Equation (23). Note that in Equation (23), $I_{M \times M}$ denotes the identity matrix, and $B_{M \times k}$ is the matrix to be filled.

$$
\begin{array}{l}
[\, I_{M \times M} \mid B_{M \times k} \,] \\
= \begin{bmatrix} 1 & 0 & \cdots & 0 & \bigg| & B_{1 \times 1} & B_{1 \times 2} & \cdots & B_{1 \times k} \\ 0 & 1 & \cdots & 0 & \bigg| & B_{2 \times 1} & B_{2 \times 2} & \cdots & B_{2 \times k} \\ \vdots & \vdots & \ddots & \vdots & \bigg| & \vdots & \vdots & \ddots & \vdots \\ 0 & 0 & \cdots & 1 & \bigg| & B_{M \times 1} & B_{M \times 2} & \cdots & B_{M \times k} \end{bmatrix} .
\end{array} \tag{23}
$$

The procedure for generating the GDA codes is shown in Equation (24), when the code rate is $M/N$.

$$
\begin{array}{c}
[\, \underset{input\ d}{I_{M \times M} \mid B_{M \times k}} \,] \rightarrow \left[ I \;\middle|\; B_{c1}^{\lfloor \frac{d}{k} \rfloor} \; B_{c2}^{\lfloor \frac{d}{k} \rfloor} \; \cdots \; B_{c(k-1)}^{\lfloor \frac{d}{k} \rfloor + 1} \; B_{ck}^{\lfloor \frac{d}{k} \rfloor + 1} \right] \\
\underset{Global\ degree\ allocation\ of\ variable\ nodes}{} \\
\downarrow \\
[\, \underset{GDAcodes}{I_{M \times M} \mid GDA(B_{M \times k})} \,] \leftarrow \left[ I \;\middle|\; \begin{matrix} B_{r1}^{\lfloor \frac{d}{M} \rfloor} \\ B_{r2}^{\lfloor \frac{d}{M} \rfloor} \\ \vdots \\ B_{r(M-1)}^{\lfloor \frac{d}{M} \rfloor + 1} \\ B_{rM}^{\lfloor \frac{d}{M} \rfloor + 1} \end{matrix} \right] , \\
\underset{Global\ degree\ allocation\ of\ check\ nodes}{}
\end{array} \tag{24}
$$

where $\lfloor \cdot \rfloor$ is the floor function, $\left\lfloor \frac{d}{k} \right\rfloor$ implies $V_j = \|B_{cj}\|_1 = \left\lfloor \frac{d}{k} \right\rfloor$, and $\left\lfloor \frac{d}{M} \right\rfloor$ indicates that $C_i = \|B_{ri}\|_1 + 1 = \left\lfloor \frac{d}{M} \right\rfloor + 1$.

---

**Algorithm 2** GDA algorithm

---

**Require:** M & N & d;
**Ensure:** GDA codes
 1: **for** $j = M + 1 \rightarrow N$ **do**
 2:     $V_j \leftarrow d/(N - M)$
 3:     **if** $j \geq N - d\%(N - M)$ **then**
 4:        $V_j \leftarrow V_j + 1$
 5:     **else**
 6:        $V_j \leftarrow V_j$
 7:     **end if**
 8: **end for**
    // Assign the total degree to the variable nodes
 9: **for** $i = 1 \rightarrow M$ **do**
10:     $C_i \leftarrow (d + M)/M$
11:     **if** $i \geq M - (d + M)\%M$ **then**
12:        $C_i \leftarrow C_i + 1$
13:     **else**
14:        $C_i \leftarrow C_i$
15:     **end if**
16: **end for**
    // Assign the total degree to the check nodes
17: **for** $j = M + 1 \rightarrow N$ **do**
18:     $C_{xj} \leftarrow V_j/M$
19:     **if** $++x \geq M - V_j\%M$ **then**
20:        $C_{xj} \leftarrow C_{xj} + 1$
       // Until the degree of the $j$th variable node is filled
21:     **else**
22:        $C_{xj} \leftarrow C_{xj}$
23:     **end if**
    // Traverse filling; the next filling position begins at an adjacent position
24: **end for**
25: **return** GDA codes

---

For example, if we take $M = 3$, $N = 5$, $d = 13$, then the obtained GDA code is

$$\underbrace{\left[ \begin{array}{c|c} I_{3\times3} & B^{13}_{3\times2} \end{array} \right]}_{input\ 13} \rightarrow \underbrace{\left[ \begin{array}{c|cc} I_{3\times3} & B^{\left\lfloor \frac{13}{2} \right\rfloor}_{c1} & B^{\left\lfloor \frac{13}{2} \right\rfloor+1}_{c2} \end{array} \right]}_{Global\ degree\ allocation\ of\ variable\ nodes}$$

$$\downarrow$$

$$\underbrace{\left[ \begin{array}{c} 1\ \ 0\ \ 0\ \ 2\ \ 2 \\ 0\ \ 1\ \ 0\ \ 2\ \ 2 \\ 0\ \ 0\ \ 1\ \ 2\ \ 3 \end{array} \right]}_{GDAcode} \leftarrow \underbrace{\left[ \begin{array}{c|c} I_{M\times M} & \begin{array}{c} B^{\left\lfloor \frac{13}{3} \right\rfloor}_{r1} \\ B^{\left\lfloor \frac{13}{3} \right\rfloor}_{r2} \\ B^{\left\lfloor \frac{13}{3} \right\rfloor+1}_{r3} \end{array} \end{array} \right]}_{Global\ degree\ allocation\ of\ check\ nodes} . \tag{25}$$

Since the lossy compression system using the P-LDPC codes is the same as [11], both the encoding and decoding complexities are linear. The complexity of Algorithms 1 and 2 is mainly related to the length and width of the protomatrices. The CA algorithm has a complexity of $O(N - M)$; the complexity of the CVA algorithm amounts to $O(N + M - 1)$; the complexity of GDA algorithm is $O(2N - M)$.

## 4. Simulation Results and Analysis

All parity check matrices $H$ were obtained by extending the aforementioned protomatrices to length $N = 2520$ in this paper. The RBP algorithm was used as the quantization algorithm, and the parameters of the RBP algorithm used for each code rate were taken from [11]. For clarity, these parameters ($RBP : 1 - \gamma, L$) are given for each figure. The simulation results over AWGN channels with [11] codes are shown in Figure 4.

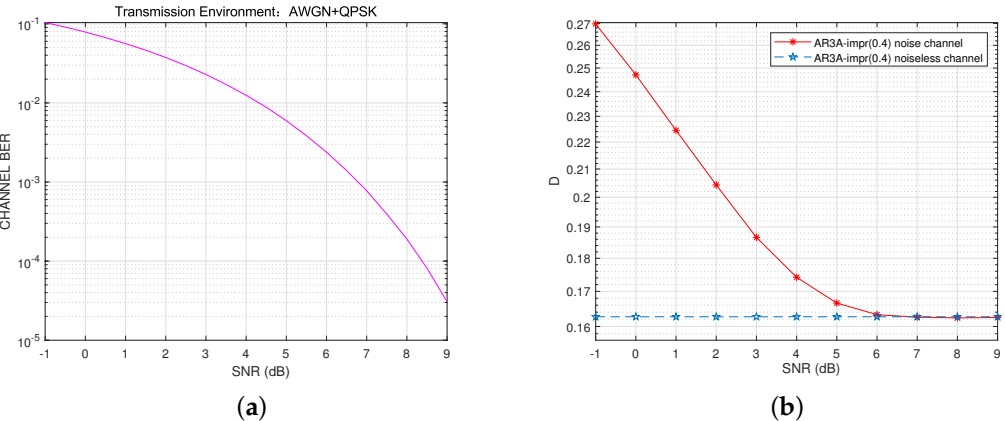

**Figure 4.** Index transmission environment and compression performance of AR3A-impr codes [11] over noisy channels. (**a**) QPSK modulation over AWGN channels (**b**) Code rate: 0.4 (RBP: 0.94 1.8).

Figure 5 shows the impacts of the total degree of the protomatrix on $D_C$, which are consistent with the theoretical derivation. It can be seen that the increase of the total degree seriously affects the compression performance in the low SNR region.

The compression performance of the codes obtained by Algorithm 1 is shown in Figures 6 and 7. Compared with the existing codes of [11], the distortion is smaller in the high SNR region.

Note that a similar allocation algorithm can also be obtained by changing the order of allocation in the CVA algorithm. If the degrees of variable nodes are reassigned first, the VCA codes can be obtained. The compression performances of the codes are also shown in Figures 6 and 7. The CVA codes may obtain better compression performance than the CA codes in the high SNR region, but larger distortion may occur in the low SNR region. The VCA codes will reduce the high distortion problem of the CVA codes in the low SNR region, but the compression performance is not as good as the CVA codes in the high SNR region.

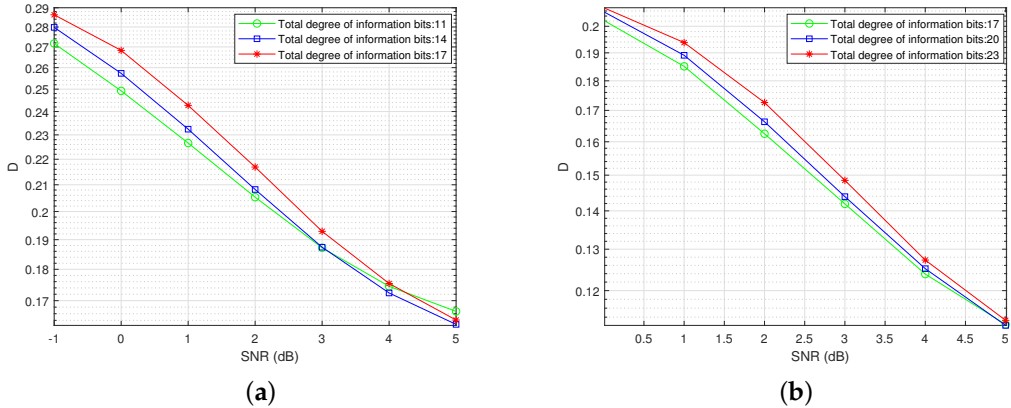

**Figure 5.** The influence of the total degree of the variable nodes corresponding to the information bit. (**a**) Code rate: 0.4 (RBP: 0.94 1.8) (**b**) Code rate: 0.57 (RBP: 0.85 2.0).

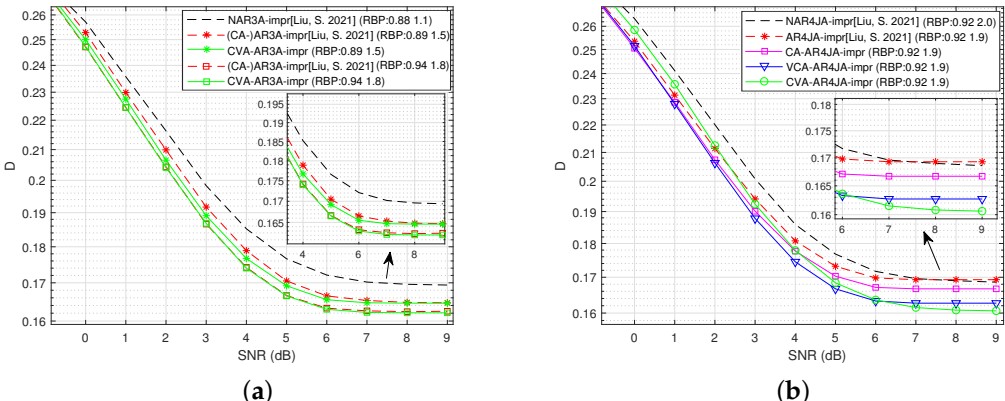

**Figure 6.** The performance of the optimized codes for lossy compression over AWGN channels (code rate: 0.4). (**a**) Based on AR3A codes [11] (**b**) Based on AR4JA codes [11].

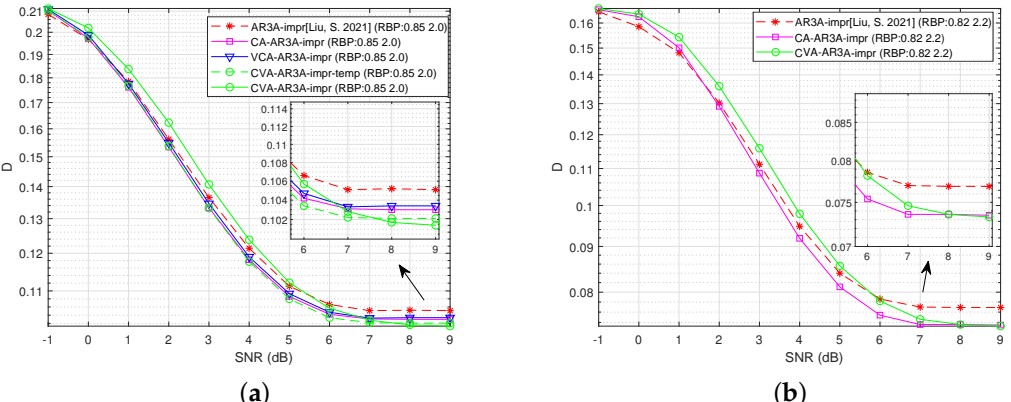

**Figure 7.** The performance of the optimized codes based on AR3A codes [11] for lossy compression over AWGN channels. (**a**) Code rate: 0.57 (**b**) Code rate: 0.66.

Figure 8 compares the distortion rate performance of GDA codes with different total degrees. It can be seen that the protomatrices generated by the GDA algorithm with different total degrees will have good compression performance in the range of the degrees of the protomatrices. The codes having better compression performance than the CVA-AR3A-impr codes can be found using the GDA algorithm. We point out that the GDA algorithm can be used to select the optimal degrees to generate protomatrices of the source codes for noise-free channels. For example, we can select a GDA code of rate 0.4 with a good compression performance, as shown in Equation (26).

$$
GDA_{3\times5}(17) = \begin{bmatrix} 1 & 0 & 0 & 2 & 3 \\ 0 & 1 & 0 & 3 & 3 \\ 0 & 0 & 1 & 3 & 3 \end{bmatrix}. \tag{26}
$$

The validity of lossy compression is measured by whether the achieved distortion is close to the theoretical distortion rate limit. Based on the analysis and the simulation results, the influence of the total degree on $D_C$ and $D_S$ can be summarized, for which the increase of total degree of **H** matrix will cause the increase of $D_C$. In order to adapt to the fluctuation of the channel environment, we can select the GDA codes having enough good compression performance, but with less degree as the source codes to realize the robustness of the channel noise and, finally, achieve the minimum average distortion in a narrow band of the SNR range. Figure 9 shows the simulation results.

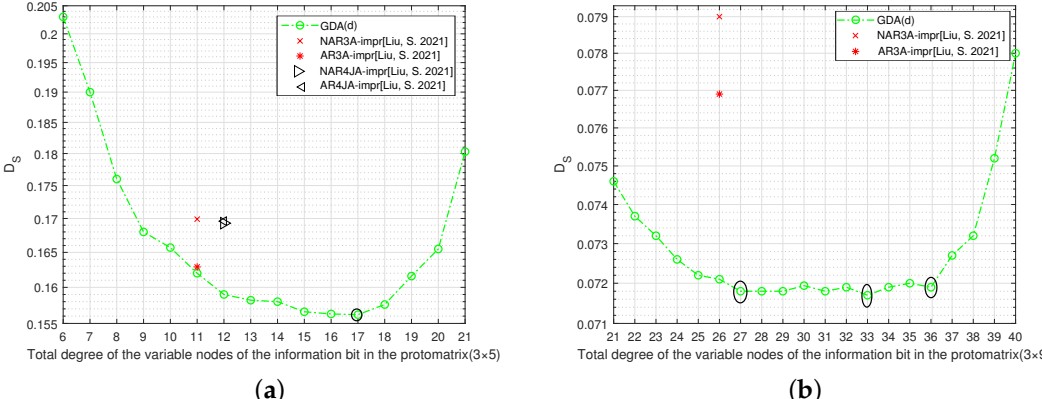

**Figure 8.** Compression performance of codes generated by different total degrees of the variable nodes corresponding to the information bit in the protomatrix using the GDA algorithm for noise-free channels [11]. (**a**) Code rate: 0.4 (**b**) Code rate: 0.66.

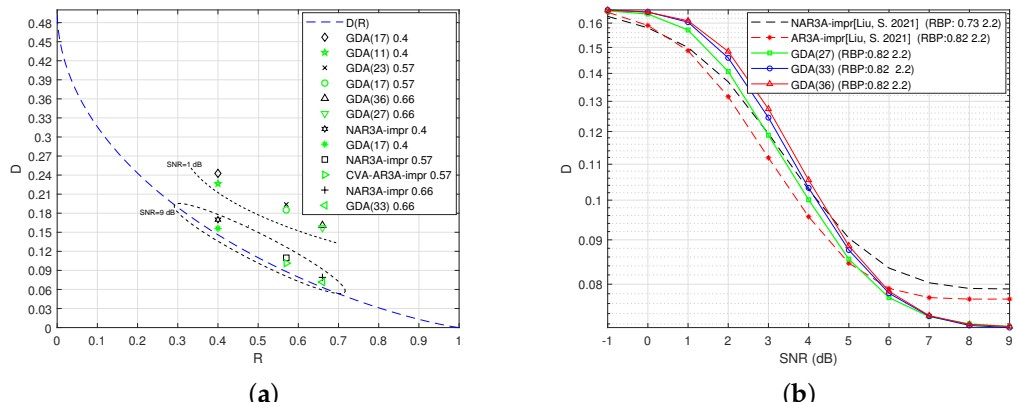

**Figure 9.** (**a**) Distortion rate performance [11]. (**b**) The GDA codes generated by different degrees have different $D$ (code rate: 0.66) [11].

For example, although $GDA_{3\times 9}(33)$ has the smallest $D_S$ at a 0.66 code rate, considering the influence of $D_C$, we can select the appropriate degree to obtain protomatrix ($d = 27$) with a 0.66 ($3 \times 9$) code rate using the GDA algorithm from Figure 8 to reduce the total distortion. The generation procedure follows the process shown in Equation (24), and the filling process is achieved by using cyclic filling, which can satisfy the conditions of global degree allocation with fewer allocation times.

$$GDA_{3\times 9}(27) = \begin{bmatrix} 1 & 0 & 0 & 1 & 1 & 2 & 1 & 2 & 2 \\ 0 & 1 & 0 & 1 & 2 & 1 & 2 & 1 & 2 \\ 0 & 0 & 1 & 2 & 1 & 1 & 2 & 2 & 1 \end{bmatrix}. \tag{27}$$

## 5. Conclusions

This paper studied the lossy compression system of equal-probability Bernoulli sources using binary P-LDPC codes over AWGN channels. The index is associated with the corresponding quantization subspace in linear block codes in this paper. Through theoretical derivation, the key factor affecting the $D_C$ of the compression system was found, i.e., a low-weight parity check matrix is highly preferred for reducing $D_C$. Through experiments, it was verified that the total degree has a notable influence on $D_C$ in the low SNR region. Moreover, the quantization subspace composed of the **H** matrix is more suitable for the source statistics by assigning the degree of variable nodes; two algorithms that optimize the existing source codes were proposed; the resultant codes further reduce $D_S$. In addition, the proposed GDA algorithm globally assigns each given degree to obtain a series of codes with

good compression performance, and it was found explicitly that using the GDA algorithm within a range of total degrees can reduce $D_S$ significantly. By selecting a smaller total degree and using the GDA algorithm to generate codes, a trade-off between compression performance and $D_C$ can be achieved. In more practical transmission environments such as multipath fading channels and shadowing channels or with multiple noises, the robustness of the compression system will be further investigated in a future work.

**Author Contributions:** Writing—original draft, R.W.; data curation, R.W.; formal analysis, R.W. and H.W.; software, R.W. and S.L.; investigation, S.L.; supervision, S.L.; writing—review and editing, H.W.; validation, H.W.; funding acquisition, L.W.; methodology, L.W.; project administrator, L.W.; resources, L.W. All authors have read and agreed to the published version of the manuscript.

**Funding:** National Natural Science Foundation of China: 61671395.

**Data Availability Statement:** Not applicable.

**Conflicts of Interest:** The authors declare no conflict of interest.

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
