# Peer review of "The Efficient Design of Lossy P-LDPC Codes over AWGN Channels"

_electronics, doi:10.3390/electronics11203337_

Round 1
Reviewer 1 Report
1. Validation of the simulation results should be proved.
2. Comparison to the previously published research works in this field of research is missed.
3. Lack of a complexity study in terms of the complexity order or computational complexity is obvious.
4. AWGN channels can be used for cable or optical fiber applications or special cases of line-of-sight propagation in wireless communications, but for IoT scenarios that you mentioned in the abstract, it may have deviated. Please take a look at this fact and do some simulations for realistic channels considering multipath fading and shadowing, LOS as well as NLOS.
5. In all figures, SNR is in dB.
Reviewer 2 Report
1. For Section-4, "Simulated Results and Analysis" is more appropiate if there was not practical measurement, test, or other mean of hardware evaluation etc.
2. Please provide details how to come to (27).
3. Some text spacing, formating, English proof read are required.
Round 2
Reviewer 1 Report
Most of the reviewers' comments are addressed or fixed in the revised version of the paper. Just the following comment has no acceptable action by the authors. It needs more attention in this paper. It can be postponed for future work but it should be mentioned in the conclusion part.
Comment: AWGN channels can be used for cable or optical fiber applications or special cases of line-of-sight propagation in wireless communications, but for IoT scenarios that you mentioned in the abstract, it may have deviated. Please take a look at this fact and do some simulations for realistic channels considering multipath fading and shadowing, LOS as well as NLOS.
